# Predictors of the Development of Protracted Bacterial Bronchitis following Presentation to Healthcare for an Acute Respiratory Illness with Cough: Analysis of Three Cohort Studies

**DOI:** 10.3390/jcm10245735

**Published:** 2021-12-07

**Authors:** Kerry-Ann F. O’Grady, Juliana Mahon, Daniel Arnold, Keith Grimwood, Kerry K. Hall, Vikas Goyal, Julie M. Marchant, Natalie Phillips, Jason Acworth, Alex King, Mark Scott, Anne B. Chang

**Affiliations:** 1Australian Centre for Health Services Innovation, Centre for Healthcare Transformation, Queensland University of Technology, Brisbane, QLD 4059, Australia; d5.arnold@qut.edu.au (D.A.); Vikas.Goyal@health.qld.gov.au (V.G.); jm.marchant@qut.edu.au (J.M.M.); anne.chang@menzies.edu.au (A.B.C.); 2School of Public Health & Social Work, Queensland University of Technology, Brisbane, QLD 4059, Australia; juliana.mahon@connect.qut.edu.au; 3NHMRC Centre for Research Excellence in Paediatric Bronchiectasis, School of Medicine and Dentistry, Menzies Health Institute Queensland, Griffith University, Gold Coast, QLD 4222, Australia; k.grimwood@griffith.edu.au; 4Departments of Infectious Disease and Paediatrics, Gold Coast Health, Gold Coast, QLD 4215, Australia; 5First Peoples Health Unit, School of Medicine and Dentistry, Griffith University, Gold Coast, QLD 4215, Australia; kerry.hall@griffith.edu.au; 6NHMRC Centre for Research Excellence in Paediatric Bronchiectasis, Department of Respiratory Medicine, Queensland Children’s Hospital, Brisbane, QLD 4101, Australia; 7Department of Emergency Medicine, Queensland Children’s Hospital, Brisbane, QLD 4101, Australia; Natalie.Phillips@health.qld.gov.au (N.P.); Jason.Acworth@health.qld.gov.au (J.A.); 8Child Health Research Centre, University of Queensland, Brisbane, QLD 4101, Australia; 9Faculty of Medicine, University of Queensland, Brisbane, QLD 4072, Australia; 10Department of Emergency Medicine, The Toowoomba Hospital, Toowoomba, QLD 4350, Australia; Alex.King@health.qld.gov.au; 11Department of Emergency Medicine, Caboolture Hospital, Caboolture, QLD 4510, Australia; Mark.Scott@health.qld.gov.au; 12Menzies School of Health Research, Charles Darwin University, Tiwi, NT 0810, Australia

**Keywords:** protracted bacterial bronchitis, children, risk factors

## Abstract

We describe the prevalence and risk factors for protracted bacterial bronchitis (PBB) following healthcare presentation for an acute cough illness in children. Data from three studies of the development of chronic cough (CC) in children were combined. PBB was defined as a wet cough of at least 4-weeks duration with no identified specific cause of cough that resolved following 2–4 weeks of appropriate antibiotics. Anterior nasal swabs were tested for 17 viruses and bacteria by polymerase chain reaction. The study included 903 children. Childcare attendance (adjusted relative risk (aRR) = 2.32, 95% CI 1.48–3.63), prior history of chronic cough (aRR = 2.63, 95% CI 1.72–4.01) and age <2-years (<12-months: aRR = 4.31, 95% CI 1.42–13.10; 12-<24 months: aRR = 2.00, 95% CI 1.35–2.96) increased risk of PBB. Baseline diagnoses of asthma/reactive airways disease (aRR = 0.30, 95% CI 0.26–0.35) or bronchiolitis (aRR = 0.15, 95% CI 0.06–0.38) decreased risk. *M. catarrhalis* was the most common organism (52.4%) identified in all children (PBB = 72.1%; no PBB = 50.2%, *p* < 0.001). We provide the first data on risks for PBB in children following acute illness and a hypothesis for studies to further investigate the relationship with wheeze-related illnesses. Clinicians and parents/guardians should be aware of these risks and seek early review if a wet cough lasting more than 4-weeks develops the post-acute illness.

## 1. Introduction

Since first recognized as a distinct diagnostic entity in 2006 [1], protracted bacterial bronchitis (PBB) is internationally recognized as a significant disease in children associated with a considerable illness burden [2]. PBB is characterized by a wet cough that persists for at least 4-weeks in the absence of specific cough pointers and resolves with a 2- or 4-week course of appropriate antibiotics [3]. PBB, especially when recurrent, is considered part of a spectrum of lung disease, which untreated could progress to bronchiectasis [3,4].

While PBB-specific knowledge has been growing, much of what is now known has been garnered from children referred to specialist pediatric pulmonologists because of chronic cough with scarce community-based data. The European Respiratory Society (ERS) identified a need to better understand the natural history of PBB and collect more data on disease burden at the community level [2]. In particular, there are no data on the risk factors for the development of PBB in children following an acute respiratory illness.

Here, our primary objective was to describe the key epidemiological and clinical predictors of PBB following an acute respiratory illness with cough (ARIwC) in children aged <15-years in urban/regional communities in South-East Queensland, Australia. We hypothesized that risk factors for the development of PBB could be identified. A secondary objective was to describe the prevalence of upper airway viruses and bacteria present at the time of acute illness and at the time of diagnosis of PBB.

## 2. Materials and Methods

### 2.1. Design

This was a secondary analysis of data collected in three prospective cohort studies with recruitment undertaken between 11 December 2011 and 6 October 2018. The protocols and primary findings of each study have been published previously [5,6,7,8,9,10]. All three studies used identical methods and data collection to follow for 4-weeks children aged <15-years with an ARIwC after their presentation to either a primary healthcare clinic (PHC) or an emergency department (ED).

### 2.2. Setting

Recruitment occurred in three hospital EDs and three PHCs from regional and urban areas in South East Queensland, Australia. The hospitals included a tertiary pediatric hospital in Brisbane, a general hospital located in an outer suburb of Brisbane (Caboolture), and another general hospital in a regional area (Toowoomba). The PHCs included one located in outer Brisbane (Caboolture) and two located in regional areas (Toowoomba/Warwick) approximately 2 h west of Brisbane. The PHCs served predominantly Aboriginal and Torres Strait Islander people.

### 2.3. Participants

In studies-1 [5] and -2 [9], children were eligible for inclusion if they were aged <15-years and presented with an ARIwC. Exclusion criteria included: (i) known history of chronic lung disease (other than asthma), (ii) primary or secondary immunodeficiency, including immunosuppressive therapy, but excluding short-course (<2-weeks) oral and ongoing maintenance inhaled corticosteroids, in the 30-days prior to presentation, (iii) current or planned participation in another study during the follow-up period for these studies and (iv) insufficient English that would inhibit workbook completion and inability to provide informed consent. Study 3 [7] was a cohort study of children aged <5-years registered with a PHC and followed for 12-months to determine the incidence and outcomes of ARIwC. For this analysis, only Study 3 children who had a cough illness at baseline were included, and subsequent illnesses they had over the following 12-months were excluded.

For this analysis, only children presenting with a cough duration of <14-days at the time of enrolment and who had a known cough outcome at the end of 4-weeks of follow-up were included. Those who were diagnosed with bronchiectasis or aspiration in the respective studies were excluded.

### 2.4. Data Collection

Data were collected at baseline from medical records and via parent/guardian interview, at weekly follow-ups (by email, phone and/or in person with parents/carers) for 4-weeks, and at the time of study clinician review if a child had a persistent cough at day-28 following enrolment. Comprehensive epidemiological and clinical data, including cough history, were collected at each time point. Diagnoses at discharge were those recorded by the attending clinician in the ED or PHC and were not standardized for the study. Three attempts were made to contact parents/carers. If there was no contact for two consecutive weeks, the child was classified as lost to follow-up. Anterior nasal swabs (included swabbing of both nares) were collected at baseline and at the time of physician review.

Persistent cough was defined as the presence of daily cough over the 4-week period with no more than a 3-day/night break in cough. Children with a break in cough at any time point were classified as cough resolved, and children for whom cough status was unknown at any time point were classified as unknown cough status unless at a previous timepoint the cough had resolved.

On day-28, children with persistent cough were invited for clinical review by a pediatric respiratory physician within 2-weeks of this timepoint. An exception was for some children in study 2 [10], where those with persistent cough were randomized to either an intervention arm and reviewed within the next 2-weeks or to a control arm and reviewed after day-56 if their cough persisted [10]. Children were reviewed in accordance with cough management guidelines [11,12] and in Study 2, according to a cough management algorithm [13].

### 2.5. Primary Outcome

The primary outcome was a diagnosis of PBB by study clinicians at review. For a provisional diagnosis of PBB, the child had to have at least 4-weeks of wet cough in the absence of specific cough pointers. The 4-weeks of persistent wet cough were confirmed by review of cough type recorded at weekly contacts up to day-28 and at the time of clinician review and the clinician’s history taking with the child’s parent/carer. Children with a provisional diagnosis of PBB were subsequently treated with at least 2-weeks of antibiotics (usually amoxicillin-clavulanate unless this was contraindicated because of prior hypersensitivity). The children were first prescribed a 2-week course and, if the cough did not resolve, a further 2 weeks was dispensed. The diagnosis was not confirmed until the cough had resolved following either 2- or 4-weeks of appropriate antibiotics, in accordance with the definition of PBB [3]. 

### 2.6. Laboratory Methods

All nasal swabs were processed using identical, previously published methods at the Queensland Paediatric Infectious Diseases Laboratory [5,9]. In study 2, the collection of nasal specimens was ceased approximately halfway through the study for financial reasons. Multiplex polymerase chain reaction (PCR) assays were used to detect *Streptococcus pneumoniae*, *Haemophilus influenzae* (capsulated and unencapsulated), *Moraxella catarrhalis*, *Chlamydia pneumoniae*, *Mycoplasma pneumoniae*, *Bordetella pertussis*, human rhinovirus, adenovirus, enterovirus, influenza viruses (A and B), respiratory syncytial virus (A and B), parainfluenza virus (1, 2 and 3), human metapneumovirus, human bocavirus-1, human coronaviruses (HUK1, OC43, NL63, and 229E), and human polyomaviruses KI and WU.

### 2.7. Statistical Methods

The baseline characteristics of children were described with means or medians for continuous data, and proportions for categorical data and groups were compared with the Student t-test, the Kruskal-Wallis test, or chi^2^ tests, respectively. Backward stepwise logistic regression with robust standard errors to adjust for clustering by study site was undertaken to identify independent predictors of PBB. Variables with a *p*-value < 0.10 in univariate analyses, and those considered potentially clinically relevant, were included in the models with both crude and adjusted relative risks (aRR) presented with their 95% confidence intervals (CI). Interaction terms were included in regression models where indicated. Age, sex, and First Nations status were retained as adjustment factors in all models irrespective of statistical significance.

## 3. Results

Between 11 December 2011 and 6 October 2018, 1458 participants were enrolled across the three studies, and 910 (62.1%) children were included initially in this analysis (Figure 1). The mean age was 3.2-years (standard deviation [SD] 3.1), 59.4% were male, and 16.3% identified as Australian First Nations. The most common diagnoses at baseline for the whole cohort were upper respiratory tract infections (30.9%) and asthma/reactive airways disease (25.6%); 27.3% of children had received antibiotics in the 7-days prior to presentation or at the time of presentation.

Persistent cough at day-28 following presentation was present in 254 children (27.9%, 95% CI 25.0–30.8), and 162 (63.7%) of those children completed clinician review. Reasons for not attending review were: lost to follow-up after day-28 (*n* = 32), parent decision not to attend (*n* = 26), cough resolved by time of appointment (*n* = 25) and other reasons (*n* = 9). Among those who completed their review, five were diagnosed with bronchiectasis and two with aspiration. They were subsequently excluded leaving 903 children for the final analysis (Figure 1). Children with a persistent cough for whom clinician review was not completed were classified as not having PBB. There were no differences in age, sex, Aboriginal and/or Torres Strait Islander status, and cough severity at day-28 between children who did and did not present for review.

Eighty-six children (9.5%) had a confirmed diagnosis of PBB. The median age of children with PBB was 1.0 year (interquartile range (IQR), 0.7–2.0), 59.1% were male, and 20.9% identified as Aboriginal and/or Torres Strait Islander. The baseline characteristics of children with and without PBB are presented in Table 1. 

Table 2 presents the independent predictors of PBB identified in regression models. Childcare attendance (aRR 2.32, 95% CI 1.48–3.63), a prior history of chronic cough (aRR 2.63, 95% CI 1.72–4.01), age <2-years (<12-months: aRR 4.31, 95% CI 1.42–13.1; 12-<24-months: aRR 2.00, 95% CI 1.35–2.96) and parent-reported wheeze in the past 12-months (aRR 1.89, 95% CI 1.61–2.23 were associated with an increased risk of PBB. Baseline reports of a diagnosis of asthma in the previous 12-months (aRR 0.28, 95% CI 0.24–0.33) were associated with a decreased risk. Among the baseline clinical diagnoses, asthma/reactive airways disease (aRR 0.32, 95% CI 0.15–0.67) and bronchiolitis (aRR 0.14, 95% CI 0.06–0.31) were associated with a decreased risk of PBB. Eight of 231 children with asthma/RAD at baseline were diagnosed with PBB. 

Baseline nasal swabs were available for PCR testing for 687/903 (76.1%) children, 68/86 (79.1%) children diagnosed with PBB, and 619/817 (75.8%) without PBB (Table 3). Overall, both viruses and bacteria were detected in 48.8% of swabs, 55.9% in children with PBB, and 48.0% in those without this diagnosis (*p* = 0.216). Co-detection of ≥2 viruses and/or bacteria occurred in 68.2% of swabs, 79.3% in children with PBB and 67.1% of children without PBB (*p* = 0.040). The most common bacterium identified at baseline was *M. catarrhalis* (52.4%), with a higher proportion detected in children subsequently diagnosed with PBB (72.1%) compared to 50.2% in other children (*p* < 0.001). Rhinovirus was the most common virus identified (27.9%), with no differences between children with and without PBB.

Specimens were available at both baseline and specialist review for 60 (71.4%) children with PBB (Table 4); *M. catarrhalis* was the most common organism detected at both timepoints.

## 4. Discussion

Our prospective study involving 903 children recruited from the community who presented with ARIwC determined the proportion and characteristics of, and risk factors for, children who subsequently developed PBB diagnosed by pediatric pulmonologists in accordance with cough management guidelines [11,12,13]. Overall, 9.3% of all children were diagnosed with PBB. Childcare attendance, age <2-years, parent-reported wheeze in the past 12-months and a prior history of chronic cough were associated with an increased risk of PBB, whereas a prior diagnosis of asthma and being diagnosed at the time of acute presentation with asthma/reactive airways disease or bronchiolitis were associated with a decreased risk. Upper airway detection of *M. catarrhalis* was more common in children with PBB than those without this diagnosis.

Studies examining risk factors for chronic cough generally in children post-ARIwC are limited. To our knowledge, this is the first study to examine risk factors for the development of PBB post-ARIwC in children. Furthermore, it is important as it fulfills a clinical-research gap identified by the ERS PBB task force [2]. PBB was the most common diagnosis amongst all children reviewed by clinicians following identification of chronic cough post-ARIwC in the three primary studies on which this analysis was based [6,8,10]. This is consistent with Australian data, where PBB is also amongst the most common diagnoses in children with established chronic cough referred to tertiary pediatric respiratory clinics, particularly in younger children [1,13,14,15,16,17,18]. An advantage of our study is most studies of chronic cough in children are based on those referred to specialist centers, often after long durations of cough [1,13,14,15,16,17,18]. Identifying, diagnosing, and treating chronic cough and PBB early in the illness improves clinical outcomes [10]. Our study enrolled children early in an ARIwC, and 22% of children diagnosed with PBB had a dry cough at baseline, suggesting an early transition from dry to wet cough that persisted. The rapid assessment and treatment of these children are important for preventing adverse sequelae and improving the quality of life of children and their carers. Additional strengths of our study are that all three included studies employed identical methods to determine cough persistence in children following acute presentation to healthcare and the inclusion of both PHC and ED sites. Furthermore, pediatric pulmonologists established PBB diagnosis following evidence-based guidelines [11,12] and algorithms [13].

Similar to the primary studies on the development of chronic cough [6,8,19], our study found that childcare attendance, young age, and a prior history of chronic cough were risk factors for PBB development. Amongst children in childcare, those who continued to attend while unwell with a cough illness were 13 times (adjusted odds ratio (OR) = 12.9, 95% CI 3.9, 43.3) more likely to develop chronic cough than those who remained absent [19]. Moreover, when children with chronic wet cough undergoing bronchoscopy were compared with children undergoing bronchoscopy for indications other than a cough, those with PBB were more likely to have attended childcare (OR 8.4, 95% CI 2.3–30.5), although this effect was no longer significant on multivariable analyses [20]. We also found that parent-reported wheeze in the past 12-months increased the risk of having PBB. This is not surprising as previously reported wheeze by parents is not uncommonly described in PBB cohort studies [20], although frequently absent on auscultation by a doctor [21]. Furthermore, the reliability of parent-reported wheeze varies, and disagreement with physicians and objective tests is common, particularly as an indicator for the diagnosis of asthma [22,23]. However, despite the very small numbers, when considering interaction effects with parent-reported wheeze and a parent-reported diagnosis of asthma in the past 12-months, a protective effect was observed.

While asthma can coexist with PBB and misdiagnosis of asthma in young children is common [22,24], the finding in our study that children with diagnoses of wheeze-related illnesses (asthma, reactive airways disease, and bronchiolitis) at the time of acute presentation were less likely to develop PBB contradicts the parent-reported history of wheeze as a risk factor.

Our finding of a decreased risk of PBB in children with asthma and bronchiolitis has not been reported previously. As our study did not involve mechanistic components, reasons for this finding can only be speculated. Nevertheless, a systematic review of studies on a chronic cough that assessed time-to-cough resolution found that when asthma is treated effectively, the cough resolves within 2–4 weeks [25]. Indeed, the first study that reported on adults who presented with cough found that in all seven adults (who also had airway obstruction evidenced on spirometry), their cough resolved within a week of treatment [26]. Unsurprisingly, evidence-based cough guidelines recommend that if asthma medications are used to treat cough, a trial period of 2–4 weeks suffices and should be ceased if not effective [27]. Moreover, children with acute asthma have a faster resolution of cough and subjective acute respiratory illness symptoms (measured using Canadian acute respiratory illness and flu scale [CARIFS]) than children with PBB [28]. Although children with asthma and PBB have similar CARIFS scores at baseline/day-1 (which were both significantly higher than controls with an acute respiratory illness), by day-14 CARIFS scores in the PBB group (median = 19.6, IQR = 25.8) were significantly higher than in both controls (median = 4.1, IQR 5.3) and those with asthma (median = 4.1, IQR 4) [28]. Reasons for this are unknown, but possible mechanisms include dysfunctional host responses [29,30] and lower airway bacteria and biofilms [31] in some children who develop a PBB phenotype.

Another reason is that although the doctors who evaluated these children for chronic cough were specialists and followed an evidenced-based algorithm, it is possible that they were less likely to diagnose PBB when children had asthma/RAD or bronchiolitis. However, this is highly unlikely as eight children of 231 with asthma/RAD at baseline were still diagnosed with PBB.

Caution is required when interpreting the observed associations with clinical diagnoses as, unlike the assessment for children with chronic cough, there was no systematic approach to the diagnosis of children at baseline. Moreover, there was a high proportion for whom no diagnosis was reported, including giving a diagnosis of “cough.” The problems with consistent and accurate diagnoses of viral respiratory illnesses, particularly with wheeze, are considered a result of inadequate understanding of underlying inflammatory processes and different clinical presentations across age groups [32]. Additionally, the diagnosis of asthma was not validated in our cohort, although there is difficulty in confirming asthma in children aged <5-years. However, given the importance of PBB, the potential for wheeze-associated illnesses to inform the underlying pathophysiological processes associated with the natural history of PBB warrants further investigation. 

While upper airway specimens are poor indicators of lower airway bacterial infection identified by culture of bronchoalveolar lavage (BAL) specimens in children with chronic wet cough [33,34], we described upper airway microbiology at the time of ARIwC between those who did and did not develop PBB. Viral-bacterial co-detection was common in all children, and there were few differences between children with and without PBB other than for *M. catarrhalis* and human polyomaviruses. Given the very small numbers (10 positive children only), the interpretation of the human polyomavirus finding is limited. Baseline upper airway detection of *M. catarrhalis* in our study was almost 50% higher in children who were subsequently diagnosed with PBB than in those who were not, a difference not observed with *S. pneumoniae* or *H. influenzae*. Furthermore, *M. catarrhalis* was detected at both baseline and physician review in 55% of children with PBB compared with 37% for *S. pneumoniae* and 23% for *H. influenzae*. *M. catarrhalis* colonization is common in healthy young children with prevalence varying by age and geography [35]. It is one of the three common causes of otitis media in children [36], exacerbations of COPD in adults [37,38], and is often within the top three bacteria detected in BAL specimens collected from children with PBB [39]. In previously published analyses of upper-airway specimens from Study 1 [40], children who were nasal swab positive for *M. catarrhalis* at baseline were 2.1 times (95% CI 1.3–3.1) more likely to have cough persistence at day-28 compared with those who were swab negative in multinomial regression models. *M. catarrhalis* was the only microbial difference between children with and without cough persistence; however, this may partly result from inadequate numbers for meaningful analysis of other organisms. There are limited studies on the role of *M. catarrhalis* in the natural history of chronic cough in children, and future studies should include serial testing at more regular intervals to further explore our findings.

Despite the novelty of our study, it has major limitations. Firstly, in addition to the above points relating to diagnoses at baseline and the difficulty in diagnosing young children with asthma, we included children diagnosed as ‘reactive airway disease’ with asthma [41]. We did this as we have found that the term ‘reactive airway disease’ is increasingly used by non-pulmonologists, including in children who have clear-cut asthma. Secondly, a relatively high proportion of children were excluded from being lost to follow-up, both prior to day-28 and at clinician review. However, analyses in each of the three primary studies suggested attrition occurred amongst children with shorter illness duration at baseline and, given the incentive of expedited specialist review of children with chronic cough and the burden of chronic cough and PBB [42,43,44], it is highly likely that these children did not have PBB (thus diluting our findings). Thirdly, our cohorts were heterogeneous. However, while this may be a limitation, it may also be a strength as PBB can coexist with wheeze and asthma phenotypes [45], and it represents a true-life scenario. Furthermore, as described above, caution is required with the baseline clinical diagnoses. However, the consistency of the association between illnesses characterized by wheeze and a decreased risk of PBB supports the need for further studies to investigate this relationship in detail.

Our study provides the first data on risk factors for PBB in children following ARIwC. It provides a hypothesis for studies to further investigate the relationship between wheeze-related illnesses and PBB in young children, as this may provide more understanding of the natural history and underlying pathophysiology of PBB. Identifying intervention points to prevent progression and, possibly, recurrent episodes associated with the development of bronchiectasis [45] would advance efforts to reduce the disease burden. Clinicians should be aware of the potential for PBB following ARIwC, particularly in young children with prior history of chronic cough and those in childcare. Parents/carers should be counseled to seek a timely medical review if a wet cough of >4-weeks occurs.

## Figures and Tables

**Figure 1 jcm-10-05735-f001:**
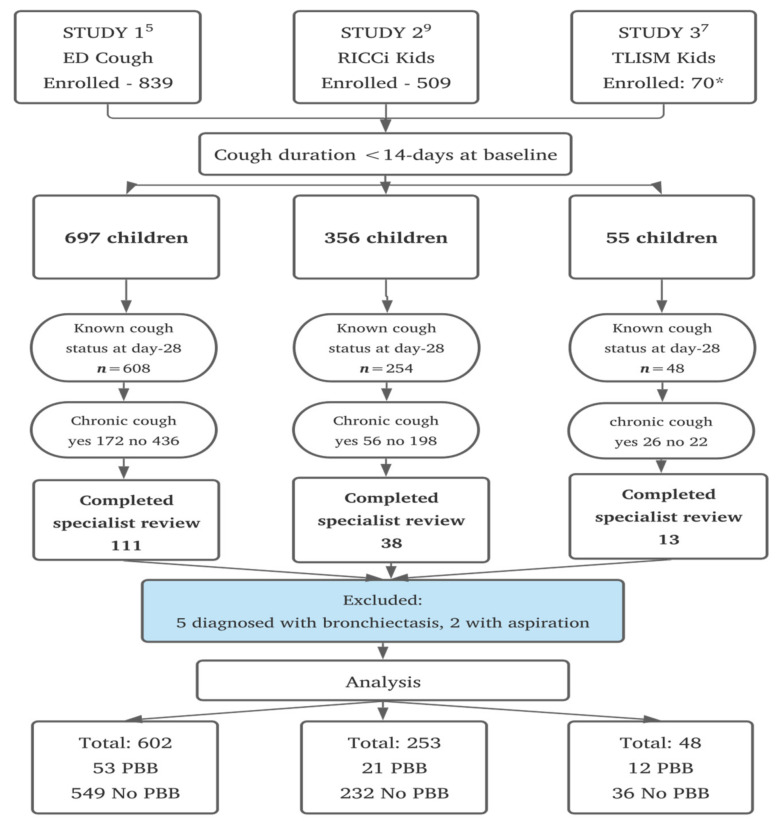
Stages to the final dataset for analysis.

**Table 1 jcm-10-05735-t001:** Baseline characteristics of children presenting with cough duration <14-days by a diagnosis of PBB following day-28 *.

Variable	PBB Diagnosis		Total	*p*-Value ^ŧ^
	No *n* = 817 (90.5%)	Yes *n* = 86 (9.5%)	*n* = 903 (100%)	
**Study**				
ED Cough	549 (67.2)	53 (61.6)	602 (66.7)	0.001
RICCi Kids	232 (28.4)	21 (24.4)	253 (28.0)	
TLSIMM Kids	36 (4.4)	12 (14.0)	48 (5.3)	
**Age group (years)**				
<1	149 (18.2)	32 (37.2)	181 (20.0)	<0.001
1-<2	208 (25.5)	31 (36.1)	239 (26.5)	
2-<5	248 (30.3)	20 (23.3)	268 (29.7)	
≥5	212 (26.0)	3 (3.5)	215 (23.8)	
Sex				
Male	482 (59.0)	52 (60.5)	534 (59.1)	0.792
Female	335 (41.0)	34 (39.5)	369 (40.9)	
**Indigenous status of the child**				
Other	689 (84.3)	68 (79.1)	757 (83.8)	0.207
First Nations Australian	128 (15.7)	18 (20.9)	146 (16.1)	
**Birthweight**				
<2500 g	72 (8.8)	6 (7.0)	78 (8.6)	0.564
≥2500 g	745 (91.2)	80 (93.0)	825(91.4)	
**Gestational age**				
<37-weeks	82 (10.0)	9 (10.5)	91 (10.1)	0.900
≥37-weeks	735 (90.0)	77 (89.5)	812 (89.9)	
**Season of enrolment**				
Summer	155 (19.0)	15 (17.4)	170 (18.8)	0.236
Autumn	271 (33.2)	22 (25.6)	293 (32.4)	
Winter	273 (33.4)	38 (44.2)	311 (34.4)	
Spring	118 (14.4)	11 (12.8)	129 (14.3)	
**Attends childcare**				
Yes	368 (45.0)	53 (61.6)	421 (46.6)	0.004
No	370 (45.3)	31 (36.1)	401 (44.4)	
Not applicable (≥6-years of age)	79 (9.7)	2 (2.3)	81 (9.0)	
**Exposed to indoor tobacco smoke**				
Yes	114 (13.9)	11 (12.8)	125 (13.8)	0.766
No	703 (86.1)	75 (87.2)	778 (86.2)	
**Number of other children in the house**				
0	230 (28.1)	36 (41.8)	266 (29.5)	0.027
1	347 (42.5)	28 (32.6)	375 (41.5)	
≥2	240 (29.4)	22 (25.6)	262 (29.0)	
**Pets at home**				
Yes	429 (52.5)	52 (60.5)	481 (53.2)	0.160
No	388 (47.5)	34 (39.5)	422 (46.7)	
**Previous history of cough lasting >4-weeks**				
Yes	160 (19.6)	29 (33.7)	189 (20.9)	0.002
No	657 (80.4)	57 (66.3)	714 (79.1)	
**History of wheeze (parent-reported)**				
Yes	447 (54.7)	49 (57.0)	496 (54.9)	0.688
No	370 (45.3)	37 (43.0)	407 (45.1)	
**History of eczema**				
Yes	207 (25.3)	13 (15.1)	220 (24.4)	0.036
No	610 (74.7)	73 (84.9)	683 (75.6)	
**Asthma diagnosis in past 12-months**				
Yes	160 (19.6)	4 (4.6)	164 (18.2)	0.001
No	657 (80.4)	82 (95.4)	739 (81.8)	
**Cough type at presentation**				
Dry	286 (35.0)	19 (22.1)	305 (33.8)	0.009
Wet	313 (38.3)	32 (37.2)	345 (38.2)	
Variable	218 (26.7)	35 (40.7)	253 (28.0)	
**Steroids (inhaled or oral) for current illness ^§^**				
Yes	277 (33.9)	17 (19.8)	294 (32.6)	0.008
No	540 (66.1)	69 (80.2)	609 (67.4)	
**Antibiotics for current illness ^§§^**				
Yes	225 (27.5)	22 (25.6)	247 (27.3)	0.698
No	592 (72.5)	64 (74.4)	656 (72.7)	
**Diagnosis at discharge**				
Upper respiratory tract infection ^£^	242 (29.6)	37(43.0)	279 (30.9)	Not done
Asthma/reactive airways disease	223 (27.3)	8 (9.3)	231 (25.6)	
Croup	81 (9.9)	9 (10.5)	90 (10.0)	
Bronchiolitis	72 (8.8)	3 (3.5)	75 (8.3)	
Pneumonia/lower respiratory tract infection	52 (6.3)	7 (8.3)	59 (6.5)	
Bronchitis	4 (0.5)	1 (1.2)	5 (0.5)	
Other ^ꝉ^	88 (10.8)	8 (9.3)	96 (10.6)	
Not documented	55 (6.7)	13 (15.1)	68 (7.5)	
**Hospitalized for current illness**				
Yes	278 (34.0)	20 (23.3)	298 (33.0)	0.043
No	539 (66.0)	66 (76.7)	605 (67.0)	

Abbreviations: ED Cough—Emergency Department Cough Study; RICCi Kids—Researching Intervention in Chronic Cough in Kids; TLSIM Kids—Tooth and Lung Sickness in Murri Kids; PBB—protracted bacterial bronchitis. ^ŧ^
*p*-value—this is either the comparison of proportions between children with and without PBB or when there are multiple categories; it is the overall p-value for this comparison across multiple strata. * Children who did not have persistent cough at day-28 and amongst those who did have persistent cough at day-28 but did not present for review or had a diagnosis 
other than PBB were classified as not having PBB. **^§^** Steroid 
use in 7-days prior to presentation or given/prescribed at presentation. **^§§^** Antibiotic use in 7-days prior to presentation or given/prescribed at 
presentation. ^£^ Upper respiratory tract infection includes cases 
with a diagnosis of unspecified viral illness/infection. ^ꝉ^ Other includes 10 
cases of influenza/ influenza-like illness and 4 cases of pertussis, none of 
whom developed PBB.

**Table 2 jcm-10-05735-t002:** Predictors for protracted bacterial bronchitis following an acute respiratory illness with cough in children aged <15-years, (*n* = 903) ^ŧ^.

Variable	PBB/*n*	Crude RR	95% CI	Adjusted RR	95% CI	*p*-Value
Female sex	34/369	1.03	0.68–1.56	0.93	0.54–1.58	0.778
Australian First Nations child	18/146	1.42	0.87–2.31	1.05	0.82–1.35	0.689
Age group (months)						
<12-months	32/181	4.30	2.41–7.66	4.31	1.42–13.1	0.010
12-<24-months	31/239	3.10	1.75–5.50	2.00	1.35–2.96	0.001
>24-months	23/483	Ref		Ref		
Childcare attendance	53/421	1.86	1.22–2.83	2.32	1.48–3.63	<0.001
Prior history of cough lasting >4-weeks	29/189	1.79	1.17–2.76	2.63	1.72–4.01	<0.001
Parent reported wheeze past 12-months	49/496	1.32	0.80–2.20	1.73	1.17–2.58	0.007
Diagnosis of asthma in past 12-months	4/164	0.25	0.10–0.61	0.55	0.27–1.13	0.105
Diagnosis of asthma and parent reported wheeze past 12-months *	2/126	0.15	0.04–0.59	0.35	0.16–0.75	0.007
History of eczema	13/220	0.55	0.31–0.98	0.73	0.48–1.11	0.144
Clinical diagnosis on the day of enrolment						
Upper respiratory tract infection ^£^	37/279			Ref		
Asthma/RAD	8/231	0.23	0.11–0.51	0.32	0.15–0.67	0.002
Bronchiolitis	3/75	0.27	0.08–0.91	0.14	0.06–0.31	<0.001
Bronchitis	1/5	1.64	0.18–15.0	3.24	0.14–73.77	0.461
Croup	9/90	0.73	0.34–1.57	0.73	0.38–1.39	0.343
Pneumonia, LRTI	7/59	0.88	0.37–2.08	1.05	0.74–1.68	0.879
Other ^ꝉ^	8/96	0.59	0.27–1.32	0.84	0.71–0.99	0.036
Not documented ^§^	13/68	1.54	0.77–3.10	1.67	0.93–3.01	0.083

Abbreviations: CI—confidence interval; LRTI—lower respiratory tract infection; PBB—protracted bacterial bronchitis; RAD—reactive airways disease; RR—relative risk. ^ŧ^ Models incorporated robust standard errors to account for clustering by study. * Interaction term for asthma x wheeze. ^£^ Upper respiratory tract infection includes cases with a diagnosis of unspecified viral illness/infection. ^ꝉ^ Other includes 10 cases of influenza/influenza-like illness and 4 cases of pertussis, none of whom developed PBB. ^§^ Not documented includes children who were given a diagnosis of “Cough”.

**Table 3 jcm-10-05735-t003:** Baseline microbiology (*n* = 687 children).

	PBB			
	No *n* = 619 (%)	Yes *n* = 68 (%)	Total *n* = 687 (%)	*p*-Value ^ŧ^
**Organism detected**				0.146
None	84 (13.6)	8 (11.8)	92 (13.4)	
Bacteria only	166 (26.8)	20 (29.4)	206 (28.0)	
Virus only	72 (11.6)	2 (2.9)	74 (10.8)	
Both virus and bacteria	297 (48.0)	38 (55.9)	335 (48.8)	
**Total bacteria detected**				0.141
0	156 (25.2)	10 (14.7)	166 (24.2)	
1	189 (30.5)	20 (29.4)	209 (30.4)	
2	169 (27.3)	25 (36.8)	194 (28.2)	
3	94 (15.2)	10 (14.7)	104 (15.1)	
4	10 (1.6)	3 (4.4)	13 (1.9)	
5	1 (0.2)	0	1 (0.1)	
**Total viruses detected**				0.169
0	151 (24.4)	12 (17.6)	163 (23.7)	
1	241 (38.9)	23 (33.8)	264 (38.4)	
2	181 (29.2)	24 (35.3)	205 (29.8)	
3	40 (6.5)	7 (10.3)	47 (6.8)	
4	6 (1.0)	2 (2.9)	8 (1.2)	
**Total viruses and/or bacteria detected**				0.015
0	81 (13.1)	8 (11.8)	89 (12.9)	
1	123 (19.9)	6 (8.8)	129 (18.8)	
2	106 (17.1)	9 (13.2)	115 (16.7)	
3	104 (16.8)	12 (17.6)	116 (16.9)	
4	105 (17.0)	23 (33.8)	128 (18.6)	
5+	100 (16.2)	10 (14.7)	110 (16.0)	
**Specific bacteria detected**				
*Streptococcus pneumoniae*	307 (49.6)	32 (47.1)	339 (49.3)	0.691
*Haemophilus influenzae*	207 (33.4)	28 (41.2)	235 (34.2)	0.198
*Moraxella catarrhalis*	311 (50.2)	49 (72.1)	360 (52.4)	<0.001
*Bordetella pertussis*	5 (0.8)	0	5 (0.7)	-
*Mycoplasma pneumoniae*	8 (1.3)	0	8 (1.2)	-
*Chlamydia pneumoniae*	3 (0.5)	0	3 (0.4)	-
**Specific viruses detected**				
Rhinovirus	172 (27.8)	20 (29.4)	192 (27.9)	0.735
Adenovirus	29 (4.7)	3 (4.4)	32 (4.6)	0.934
Respiratory syncytial virus (A & B)	104 (16.8)	7 (10.3)	111 (16.1)	0.087
Parainfluenza (1, 2 & 3)	35 (5.6)	6 (8.8)	41 (6.0)	0.283
Influenza (A & B)	22 (3.5)	2 (2.9)	24 (3.5)	1.000
Enterovirus	24 (3.9)	1 (1.5)	25 (3.6)	0.499
Human bocavirus-1	16 (2.3)	2 (2.9)	18 (2.6)	0.693
Human metapneumovirus	14 (2.0)	2 (2.9)	16 (2.3)	0.664
Human coronavirus HUK1, OC43, NL63 & 229E	18 (2.9)	3 (3.7)	21 (3.1)	0.454
Human polyomavirus: KI & WU	6 (1.0)	4 (5.9)	10 (1.5)	0.012

^ŧ^ *p*-value—for comparison of proportions between children with and without PBB. Where multiple categories exist, the p-value is for the comparison across multiple strata.

**Table 4 jcm-10-05735-t004:** Nasal microbiology in children with PBB at baseline, at time of specialist review, and who were positive at both time points (*N* = 60 children with specimens at both time points).

	Baseline	Specialist Review	Both Timepoints in the Same Children
**Organism detected**			
None	6 (10.0)	8 (13.3)	2 (3.3)
Bacteria only	16 (26.7)	25 (41.7)	7 (11.7)
Virus only	2 (3.3)	3 (5.0)	0
Both virus and bacteria	36 (60.0)	24 (40.0)	14 (23.3)
**Specific bacteria detected**			
*Streptococcus pneumoniae*	28 (46.7)	38 (63.3)	22 (36.7)
*Haemophilus influenzae*	24 (40.0)	24 (40.0)	14 (23.3)
*Moraxella catarrhalis*	44 (73.0)	43 (71.7)	33 (55.0)
*Bordetella pertussis*	0	0	0
*Mycoplasma pneumoniae*	0	0	0
*Chlamydia pneumoniae*	0	0	0
**Specific viruses detected**			
Rhinovirus	19 (31.7)	12 (20.0)	2 (3.3)
Adenovirus	3 (5.0)	6 (10.0)	0
Respiratory syncytial virus (A & B)	7 (11.7)	1 (1.7)	0
Parainfluenza (1, 2 & 3)	6 (10.0)	2 (3.3)	1 (1.7)
Influenza (A & B)	2 (3.3)	1 (1.7)	0
Enterovirus	1 (1.7)	5 (8.3)	0
Human bocavirus-1	2 (3.3)	2 (3.3)	0
Human metapneumovirus	0	0	0
Human coronavirus HKU1, OC43, NL63 & 229E	2 (3.3)	5 (8.3)	0
Human polyomavirus—KI & WU	3 (5.0)	5 (8.3)	1 (1.7)

Note: Mean number of days between baseline swab and specialist swab = 45 days (95% CI 41.8–48.5).

## Data Availability

The data from this study can be made available following a written request to the authors; data on Aboriginal and Torres Strait Islander children may be subject to some restrictions.

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
