# Peer review of "Predictors of the Development of Protracted Bacterial Bronchitis following Presentation to Healthcare for an Acute Respiratory Illness with Cough: Analysis of Three Cohort Studies"

_jcm, 2021, doi:10.3390/jcm10245735_

Round 1

Reviewer 1 Report

Overall this is an important study for pulmonologists and pediatricians, evaluating risk factors in development of PBB after viral illness.  Moreover, it is noted that the study is one of the ERS with regards to PBB.  The study design is well described, and the conclusions support the purpose and results of the study.   The discussion highlights the validity of the results and the limitations of the study.  I do have a few questions regarding the tables, specifically tables 1 and 3:

  1. In table 1, the row labeled "ED cough"--the total numbers 549+53 would equal 602.  I apologize if I missed something in the text, but the total per the table equals 604.  This would not change the outcome of the percentages or statistical significance, but it may be worthwhile to explain what happened to the other 2 patients or the total (again, sorry if I missed an explanation).  
  2. For tables 1 and 3, it may be a good idea (for clarity for the reader) to describe in the caption what the p-values are indicating or representative of.  There are multiple "rows within rows" but only 1 p-value per row. I'm assuming these values were representative of the totals, but again, some clarity in the caption may help.

Overall, a good study with results that could be clinically useful.  

Author Response

We thank the reviewer for their time in undertaking this review. Please find our responses below:

study.  I do have a few questions regarding the tables, specifically tables 1 and 3:

  1. In table 1, the row labeled "ED cough"--the total numbers 549+53 would equal 602.  I apologize if I missed something in the text, but the total per the table equals 604.  This would not change the outcome of the percentages or statistical significance, but it may be worthwhile to explain what happened to the other 2 patients or the total (again, sorry if I missed an explanation).  

Response: Thank you for picking up our error and apologies for the typo error. The correct total is 602, consistent with Figure 1. We have now corrected it in the revised manuscript

  1. For tables 1 and 3, it may be a good idea (for clarity for the reader) to describe in the caption what the p-values are indicating or representative of.  There are multiple "rows within rows" but only 1 p-value per row. I'm assuming these values were representative of the totals, but again, some clarity in the caption may help.

Response: Thank you for the suggestion and we have added a caption to the tables. The p-values for tables 1 and 3 are those for the comparisons of proportions between PBB and no PBB. Where multiple categories exist, it is the overall p-value for this comparison across multiple strata.

  1. Overall, a good study with results that could be clinically useful.  

Response: We thank Reviewer 1 for their kind comment.

Reviewer 2 Report

This and interesting study as no earlier studies have evaluated the incidence of protracted bacterial bronchitis after upper airway infection. The authors shows a relatively high incidence of PBB after airway infection, identifying risk factors. There are however some limitations that may be properly addressed.

Methods:

Inclusion criteria were children < 15 år, however few children were older than 5 y. and older children may be another clinical group with different risk factors, and the could consider excluding those > 5 y of age.   

Lost to follow-up, could there be a selection bias recovered children are more probably lost to follow up ?

A large number of children had dry cough at presentation (tbl 1), were they still assesses as having had wet cough > 4 weeks at the time of treatment for possible PBB ? Persistent cough is only defined as cough > 4 weeks in the study, whereas PBB is defined with wet cough. This should be clarified. Moreover, Even though >4 weeks of wet cough is in the definition of PBB, this is still a short duration compared to most of children normally treated as possible PBB. How was it assessed that cough for 4 weeks were not

The description of diagnosing PBB treatment is scarce. Did all children with cough > 4 weeks receive antibiotics ? How many with persistent cough were treated with AB ? How many of those treatment responded and diagnosed as PBB – did all theses had wet cough ? Were all treated with the same protocol (type of antibiotics) and how many were treated 2 and 4 weeks respectively ?

Tbl 1, percentages in columns, could that be % of those in the row having PBB or not, rather than % of those with PBB, as in the upper row !  Eg; how many children in ED cough had PBB , not how may of PPPs were in the ED cough group.

A large proportion of children received systemic or inhaled steroids for acute illness, which is not indicated in acute viral infections. This may be discussed.

The diagnosis of asthma is difficult and less objective in that age. Careful interpretation is therefore needed, and I suggest to downsize the discussion and speculation on this.

Hence, a limitation may be that the group included is not very clear (high number with steroids, heterogeneous age group etc), which could be further addressed in the discussion

Author Response

We kindly thank the reviewer for their time with this manuscript. Please find our responses to your comments below.

  1. This and interesting study as no earlier studies have evaluated the incidence of protracted bacterial bronchitis after upper airway infection. The authors shows a relatively high incidence of PBB after airway infection, identifying risk factors. There are however some limitations that may be properly addressed.

Response: Thank you for acknowledging the novelty of our study.

  1. Methods: Inclusion criteria were children < 15 år, however few children were older than 5 y. and older children may be another clinical group with different risk factors, and the could consider excluding those > 5 y of age.  

Response: Thank you for your comment and yes, the cohort was skewed towards younger children in whom presentations to healthcare for acute respiratory illnesses with cough are known to be more common. In this regard it is very similar to other published cohorts. Thus, we considered it important to retain children over the age of 5 years given chronic cough and PBB does occur in older children [AB Chang, et al. A multi-centre study on chronic cough in children: burden and etiologies based on a standardized management pathway. Chest 142 (4):943-950, 2012; Kantar et al, ERS statement on protracted bacterial bronchitis in children. ERJ 2017 50: 1602139]. We therefore ensured that the regression models accounted for the effects of age.

  1. Lost to follow-up, could there be a selection bias recovered children are more probably lost to follow up ?

Response: It is not clear if you referring to the children from the original cohorts that we excluded given cough status at day 28 was unknown, or children who did not present for physician review if they did have chronic cough at day 28. We excluded the former primarily for the reason you indicated as well as attempting to minimise the relevant selection bias. The primary studies upon which this analysis was based reported that children lost to follow-up were most likely those that had recovered, and we addressed this in lines 325-330 of the discussion in the limitations section of our original manuscript (now lines 352-357) in our revised manuscript).

  1. A large number of children had dry cough at presentation (tbl 1), were they still assesses as having had wet cough > 4 weeks at the time of treatment for possible PBB ? Persistent cough is only defined as cough > 4 weeks in the study, whereas PBB is defined with wet cough. This should be clarified. Moreover, Even though >4 weeks of wet cough is in the definition of PBB, this is still a short duration compared to most of children normally treated as possible PBB. How was it assessed that cough for 4 weeks were not.

Response: Cough type at baseline was not associated with the development of chronic cough or the diagnosis of PBB at physician review. A dry cough may rapidly change to a wet cough and we captured cough type at each of the weekly follow-ups (days 7, 14, 21, and 28) and at the time of physician review. So, at the time of physician review, these data as well as the physician’s assessment of the child with the child’s parent/carer were the basis for confirming 4-weeks of wet cough. We have added this to the manuscript in lines 123-133 of the methods under 2.5 Primary Outcome.

As described in the manuscript, the children were assessed according to published cough algorithms and guidelines where chronic cough is defined as > 4 weeks and PBB has to have at least 4 weeks of wet cough. Indeed, many children diagnosed with persistent wet cough are not assessed, diagnosed or treated for some time after the 4 weeks threshold and, as per our clinical trial assessing early intervention in chronic cough (reference 10) earlier diagnosis and intervention improves clinical outcomes. We have added text to the discussion (lines 244-249) to highlight the importance of early diagnosis

Unfortunately, there was a part of the sentence missing at the end of your comment so with apologies we were not sure what your final question was.

  1. The description of diagnosing PBB treatment is scarce. Did all children with cough > 4 weeks receive antibiotics ? How many with persistent cough were treated with AB ? How many of those treatment responded and diagnosed as PBB – did all theses had wet cough ? Were all treated with the same protocol (type of antibiotics) and how many were treated 2 and 4 weeks respectively ?

Response: We apologise for any lack of clarity. As described in the manuscript, children were assessed according to published cough management guidelines and algorithms. PBB was suspected if a child had 4 weeks of wet cough in the absence of specific cough pointers. Therefore, if a child met this criterion they were treated with antibiotics and if the cough resolved after 2 – 4 weeks of antibiotics the diagnosis was confirmed. As per our response to point 4, we have now clarified the diagnosis and treatment in lines 130-133 of the methods under section ‘2.5 Primary Outcome’.

  1. Tbl 1, percentages in columns, could that be % of those in the row having PBB or not, rather than % of those with PBB, as in the upper row !  Eg; how many children in ED coughhad PBB, not how may of PPPs were in the ED cough group.

Response. Thank you for your comment. Different people have different preferences for how data are displayed and interpreted. With respect, our preference is to leave the table unchanged.

  1. A large proportion of children received systemic or inhaled steroids for acute illness, which is not indicated in acute viral infections. This may be discussed.

Response: Thank you for your suggestion. Inappropriate use of steroids in acute respiratory illnesses in children is an important topic, which is well documented elsewhere. However, steroids were not associated with our primary outcome and while this is important, it was not the aim of our paper to look at treatment of children with acute cough. We have previously examined this in another study [S. Anderson-James, J. M. Marchant, A. B. Chang, J. P. Acworth, N. T. Phillips, B. J. Drescher, V. Goyal, and K. F. O'Grady. Burden and emergency department management of acute cough in children. J Paediatr Child Health, 2018]. Unfortunately, it is beyond the scope of this paper to address the appropriateness of treatment given to children at the time of acute illness.

  1. The diagnosis of asthma is difficult and less objective in that age. Careful interpretation is therefore needed, and I suggest to downsize the discussion and speculation on this.

Response: We agree that the diagnosis of asthma is controversial in young children and we acknowledged that in lines 272/273 of our original manuscript (now lines 282/283 in revised manuscript). We also have a paragraph on the limitations of the paper with respect to the diagnoses assigned at baseline from lines 305-316 (now lines 311-321 in revised manuscript). We have also discussed asthma in the context of wheeze related illnesses in general including bronchiolitis and reactive airways disease. As the associations between wheeze related illnesses, including but not limited to asthma, are new and strong we consider the discussion as it is currently presented is necessary to adequately address our findings.

  1. Hence, a limitation may be that the group included is not very clear (high number with steroids, heterogeneous age group etc), which could be further addressed in the discussion.

Response: We have added a sentence related to this in the study limitations part (lines 357-359) in the revised manuscript. However, based on the aims of our paper, our responses above, and reviewer 1’ comments, we respectfully feel that further discussion around these two factors is not needed. Doing so would just distract the message of our manuscript and reduce the paper’s readability and potential impact.